# Appearance-Based Gaze Estimation for Driver Monitoring

**Soodeh Nikan**      SNIKAN@FORD.COM and **Devesh Upadhyay**      DUPADHYA@FORD.COM
*Ford Motor Company, Dearborn, MI 48121*

## Abstract

Driver inattention is a leading cause of road accidents through its impact on reaction time in the face of incidents. In the case of Level-3 ($L_3$) vehicles, inattention adversely impacts the quality of driver take over and therefore the safe performance of $L_3$ vehicles. There is a high correlation between a driver's visual attention and eye movement. Gaze angle is an excellent surrogate for assessing driver attention zones, in both cabin interior and on-road scenarios. We propose appearance-based gaze estimation approaches using convolutional neural networks (CNNs) to estimate gaze angle directly from eye images and also from eye landmark coordinates. The goal is to improve learning by utilizing synthetic data with more accurate annotations. Performance analysis shows that our proposed landmark-based model, trained synthetically, is capable of predicting gaze angle in the real data with a reasonable angular error. In addition, we discuss evaluation metrics are application specific and there is a crucial requirement for a more reliable assessment metric rather than common mean angular error to measure the driver's gaze direction in $L_3$ autonomy for a control takeover request at a proper time corresponding to the driver's attention focus to avoid ambiguities.

**Keywords:** Driver attention, L3 autonomy, Takeover request, CNN-based gaze estimation

## 1. Introduction

Humans respond to environmental visual/audiotary/cognitive stimuli through eye movement Ghosh et al. (2021). Capturing eye movement patterns (gaze, fixation, saccade), as primary indicators of human intent and cognitive state, is considered beneficial in a wide range of applications such as human robot interaction, tracking driver visual attention, impairment detection, virtual reality and reading process tracking Bottos and Balasingam (2020). Driver inattention is one of the the leading causes in fatal road accidents. Inattention has significant impact in $L_3$ automated driving. As defined by Society of Automotive Engineers (SAE) International (2018), in $L_3$ autonomy, human drivers do not need to monitor the driving environment continuously and may engage in secondary tasks. During this period the autonomous agent (AG) takes over driving responsibilities. However, the AG may request the human driver to take over control in scenarios beyond beyond the capabilities of the AG. This decision of a take-over request (TOR) is derived from the automated driving system (ADS). Driver state monitoring (DSM) can explicitly assist in quantifying a driver's ability to engage in safe/efficient TOR and improve both safety and user experience. One important measure of a driver's instantaneous on/off-road attention is the gaze vector. The direction of driver's eye gaze can be quantified from the face/eye images captured by cameras inside the vehicle cabin to monitor the driver's visual attention in the driving environment. Over the last decades, with the aid of advanced computer vision and artificial intelligence (AI) technologies, automatic gaze estimation has attracted research and many methodologies have been proposed. However, achieving state-of-the-art performance for gaze prediction, in real-word scenarios, is still a challenging task due to a broad range of noise factors including head

movements, illumination variations, occlusion and low resolution and information loss, and the need for a large amounts of labelled training data. Vision based gaze tracking approaches can be classified into into two main categories. 1) Geometric model-based methods, which are quite popular in many safety-critical applications, such as automotive or virtual reality, use near infra-red illuminators with known geometry Huang et al. (2017a) to capture the corneal reflection (glint) Park et al. (2018b) and make a 3-dimensional subject-specific eye model and geometric calibration to determine point of gaze (PoG) from gaze angle. Since these methods rely on a physical model, they generalize quite easily to new subjects with little or no training data, but at the cost of dedicated hardware requirements and higher sensitivity to input noise (partial occlusions or lighting) Kellnhofer et al. (2019). Moreover, these models require calibration to recover person-specific information Cheng et al. (2021). 2) Appearance-based methods, which learn form annotated image datasets to map image information into gaze directions. They do not require additional hardware (lighting) and regress gaze directly from camera frames Kellnhofer et al. (2019). These methods are broadly divided into two subcategories. a) Conventional machine learning which extract gaze-related descriptive features from image pixels (such as histograms of oriented gradients Martinez et al. (2012)) and regress gaze angle using techniques such as support vector machine Xu et al. (2015) or random forest Huang et al. (2017b). b) Deep learning approaches that estimate direct mapping from image to the gaze vector quantitatively using neural networks (Fig 1), which is more accurate and robust against noise factors. However, compared to the superior improvements in other human modeling studies, with the aid of deep neural network (DNN) representation power, gaze estimation has not yet achieved the same level of maturity. This is primarily due to the complex eye appearance and cognitive process in the visual system, and most importantly, the lack of sufficient annotated training datasets Kellnhofer et al. (2019). In this paper we will compare the performance of appearance-based methods and assess the effect of including intermediate features (landmarks) to improve the accuracy of gaze estimation. Furthermore, in the aforementioned gaze estimation approaches, the performance is evaluated by angular error metrics which are very often defined based on special use cases relative to the screen. However, those metrics do not address relevant interpretations in specific applications such as DSM to measure the human's gaze focus and attention on the road in control transition from autonomous agent to the driver in $L_3$ autonomy. Therefore, in this work we discuss the need for contextually relevant metrics to avoid ambiguities in performance assessment, which is particularly critical in safety-related applications to reduce false positive/negative rates. The paper is organized as follows. In section 2, we will review state of the art deep learning-based gaze estimation. Section 3 and 4 will describe our dataset and appearance-based models. Section 5 evaluates the performance of the proposed models. In Sections 6 and 7 a discussion about the error metrics and their shortcomings is provided and the paper will be concluded.

## 2. Related Works

A vision-based AI framework to measure the driver's visual attention in automotive applications is shown in Fig 1. The input frames from the in-cabin camera is preprocessed and face/eye crops are fed into the AI modules to extract hand-crafted or deep descriptive features, quantify eye gaze information and map them to the region (zone) where the driver

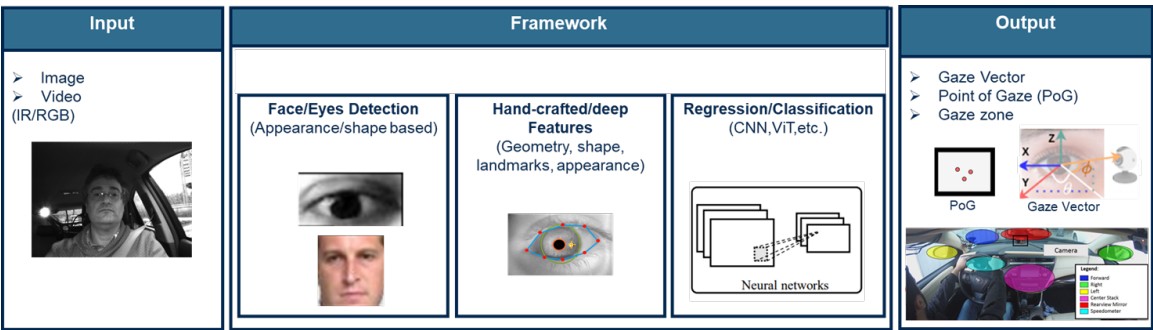

Figure 1: Vision-based AI framework for measuring the driver's visual attention in automotive

is looking. A direct mapping from face/eye appearance to the gaze vector can be learnt in deep learning-based methods without the need for extracting hand-crafted features. In 2020, Cazzato et al. Cazzato et al. (2020) presented an overview of deep learning-based gaze estimation. Recent solutions are based on different convolutional neural netwok (CNN) architectures to learn image representations and map them to gaze directions Park et al. (2018a)-Wang et al. (2018) in an end-to-end framework. The input to the network is either one eye crop Zhang et al. (2017a), face patch Zhang et al. (2017b) or the combination of both eyes and face Krafka et al. (2016). LeNet Zhang et al. (2015) was proposed by Zhang et al., as the first deep learning-based gaze estimation, where a CNN was used to extract deep features from the eye patches to estimate gaze direction. They incorporated head-pose information at the feature level to improve the accuracy. Furthermore, they extended that model to a 13-convolutional-layer network with VGG backbone as GazeNet Simonyan and Zisserman (2014) to improve the accuracy of gaze estimation. Zhang et al. Zhang et al. (2017b) applied spatial weighting on the full-face input to encode the importance of different regions. In MiNENet Perry and Fernandez (2019) the accuracy of gaze estimation was substantially improved by adopting dilated convolutions to preserve spatial resolution in the eye regions and increase the contextual information with a larger receptive field without compromising the number of parameters. In some of the recent techniques, attention mechanism has been used to weight the features from two eyes and face when a combination of all those inputs is utilized. Bao et al. Bao et al. (2021) proposed attention to combine the feature maps from two eyes and use a convolution layer to generate weights. It is not practical to train models with a datasets that covers the whole range of appearance variants among different people. Therefore, subject-invariant gaze estimation has become a research hotspot in recent years. Park et al. proposed a Pictorial Gaze approach Park et al. (2018a) using two consecutive models. The first model was Hourglass network to regress a unified representation (model of eyeball and iris) form eye crop which is then fed into a DenseNet to determine the gaze vector. Lee et al. Lee et al. (2018) proposed CycleGAN, where two generators were trained to map images from source to the target domain, and vice-versa. CycleLoss and adversarial loss were used to emphasize on the intra and inter-domain differences.

Despite significant success in the accuracy of DNN-based gaze estimation, their performance degrades in the presence of noise factors in unconstrained real-world scenarios, such as

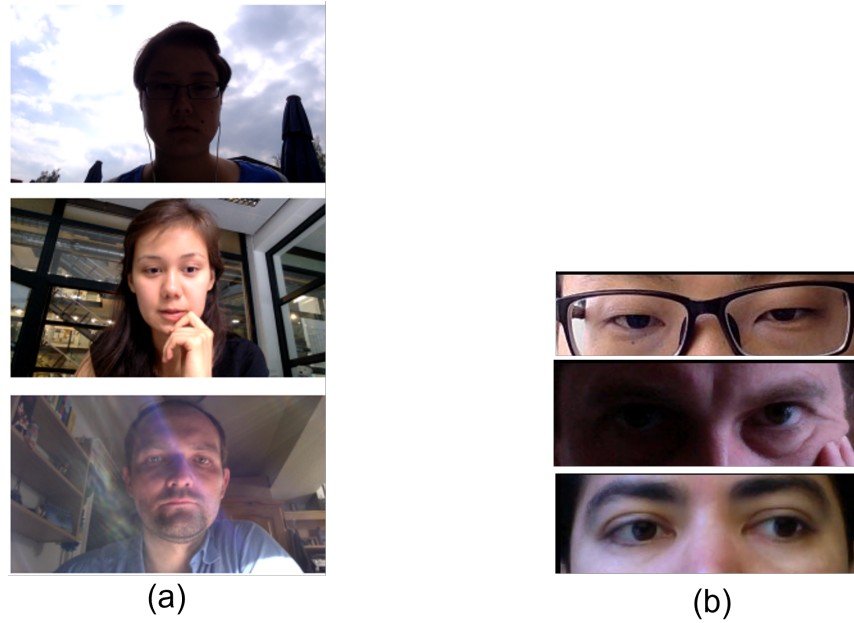

(a)  (b)

Figure 2: Sample images from MPIIGaze dataset: a) original recordings, b) cropped images in the dataset

low resolution, blur effect, harsh lighting and head-pose changes Oh et al. (2022). Moreover, they require a large-scale annotated dataset which is expensive and difficult to collect. To resolve this issue, we propose training the proposed models on high-quality synthetic eye images.

## 3. Description of the Data

In this section, we describe the real and synthetic datasets utilized to training and inference, as shown in Fig 7.

### 3.1. Real dataset - MPIIGaze

The MPIIGaze dataset, proposed by Max Planck research center Zhang et al. (2017a), contains more than 200k images that were captured in unconstrained conditions using a laptop for several months over natural everyday laptop use by 15 participants (about 30k to 1500 two-eye patches per subject) (Fig 2). In this work we utilized a subset of the dataset with an even distribution among subjects. The dataset includes 45k images of normalized 60 × 36 single eye crops (about 3000 left and right eye patches per subject) with annotations of gaze vectors. This dataset contains variability of places, time, light and shadows, as shown in Fig 2a.

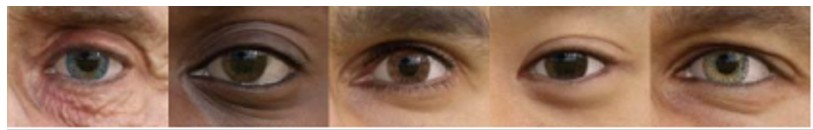

Variations in the skin tone and eye shape

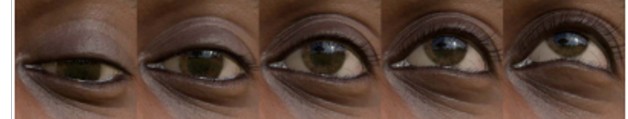

Variations in eyelid and gaze angle

Figure 3: Sample synthetic images from SynthesEyes dataset with appearance variations

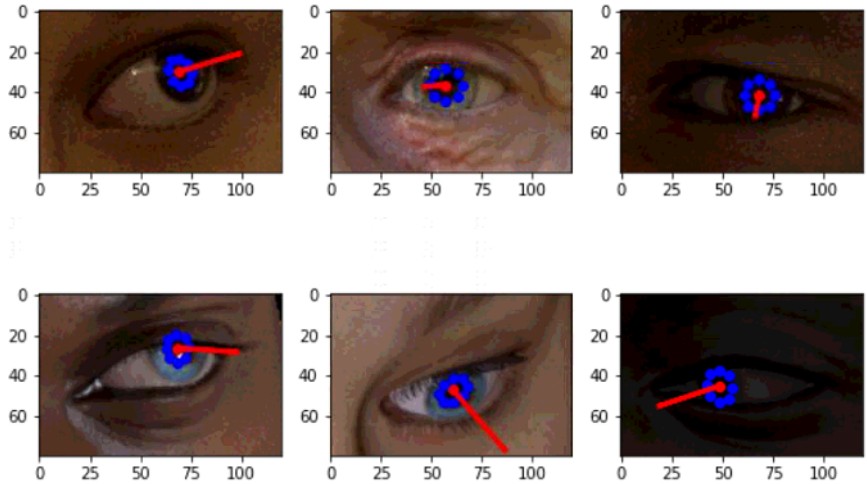

Figure 4: Examples of Gaze and landmark annotations within the SynthesEyes dataset

### 3.2. Synthetic dataset - SynthesEyes

Utilizing synthesis data, with reasonable appearance distribution, to train deep learning models is a promising solution to the issue of collecting and annotating a large-scale real dataset. SynthesEyes which was proposed by University of Cambridge Wood et al. (2015), includes 11382 synthesis images of eye region, rendered from head-scan data using a highly accurate physically based method. It contains a wide range of gaze and head directions, illumination, eye shapes, bone structures and skin colors and eyelids are posed based on gaze direction (Fig 3).The list of annotation in this dataset contains eye landmarks and gaze vector which are shown in Fig 4.

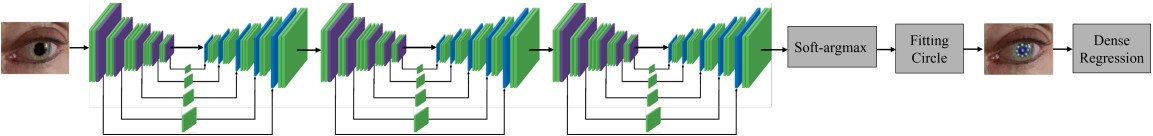

Figure 5: Block diagram of the proposed gaze regression using eye landmarks

## 4. Methodology

In this paper, we will conduct a comparison among the capabilities of various DNN models in estimating gaze direction which are trained on the real and synthetic datasets. This analysis is divided into two following strategies.

### 4.1. Strategy 1: Gaze regression directly from eye crop

In this strategy supervised regression method is proposed to infer gaze direction (yaw and pitch angles) from eye crops. Eye patches are fed to the CNN models (ResNet18, GazeNet and ResNet50) to regress gaze direction. In those models the original network structures have been used and the last layers were replaced with a fully connected regression layer with two output neurons (yaw and pitch) and the loss function was replaced by a mean squared error (MSE) cost function. The 3D gaze vector ($x, y, z$) was encoded into 2D space as pitch ($\theta$) and yaw ($\phi$), as the Euler angles between the pupil and the eyeball, as follows.

$$\theta = arcsin\,(y) \qquad , \qquad \phi = arctan\,(\frac{x}{z}) \tag{1}$$

### 4.2. Strategy 2: Gaze regression from eye landmarks

We propose an intermediate representation prior to the gaze regression by detecting eye landmarks through single eye images.The goal is to provide a robust and accurate landmark detection to improve the gaze estimation performance. Synthetic dataset provide more precised annotations even under heavy occlusion, compared to real data. Therefore, by training our model on the high-quality synthetic images, the accuracy and robustness of landmark detection will be improved which enhances the consecutive gaze estimation accordingly. To achieve that goal, we used the annotated images from SynthesEyes dataset to train a landmark detection model. In this study, a landmark detection model is utilized to detect landmark heatmaps and use those features to feed a fully-connected regression layer to estimate the gaze angle. the architecture of the proposed model is shown in Fig. 5.

Hourglass is multi-scale analysis originally proposed for human pose estimation in a downscale and upscale structure similar to auto-encoders with residuals and skip connections, and feature maps are refined multiple times at multiple scales. We utilized Hourglass model and a $1 \times 1$ convolution to generate heatmaps from detected landmark features as shown in fig. 5. Via a soft-argmax layer Honari et al. (2018) we will calculate the coordinates of the landmarks, which are the pupil contours. Then the best circle fit to the pupil's contour data points will be estimated to find the centre of the circle as the eye pupil added to the eye feature stack. In order to estimate the gaze vector, a fully connected layer was added to regress gaze direction (yaw and pitch angles) from detected eye landmarks.

## 5. Results and Evaluations

In this section, we describe the experiments conducted in this study. Models were trained for 100 epochs, $MSE$ was utilized as the loss function and Adam was the optimizer. To evaluate the performance of the proposed models, we used the mean gaze angle error as a common metric for gaze estimation problems. The gaze angle error is the cosine distance between the ground truth gaze angle and the predicted gaze angle ($g$ and $\hat{g}$, respectively) as follows.

$$\zeta = (\frac{1}{N}) \sum (\frac{180}{\pi}) \; arccos \; (\frac{g \cdot \hat{g}}{\| g \| \| \hat{g} \|}) \tag{2}$$

To assess the performance of strategy 1, the gaze estimation directly from eye image, we tested the models on the MPIIGaze dataset (Fig 2). The normalized Left and right half of the images in the dataset , were used as single eye crops. We had two separate test procedures as follows. a) In a 15 fold cross validation, every time, we excluded the images of 1 subject in MPIIGaze as the test samples and trained the models using the images of 14 other participants. Figure 6, illustrates the comparison among three models for each test set. The average cross validation errors for 15 separate test sets are shown in the second column in Table 1. b) In order to evaluate the effect of utilizing synthetic data as a remedy for the data requirement roadblock and having access to more accurate annotations, all 11382 images in SynthesEyes dataset (Fig 3) with gaze 3D vector annotation were used for training three models in strategy 1. The gaze prediction performance was tested on the images of every individual participants in the real dataset (MPIIGaze) as 15 separate test sets. The average angle errors are compared in the third column in Table 1. As shown in the table, the performance degrades drastically by training the models synthetically and the angle errors become larger (in the second column). One reason is due to different settings in target (real) dataset and the synthetic training samples. The models cannot capture all variations in the test set as shown in Fig 7. We anticipate similar problems in utilizing these gaze estimation models in automotive and in-vehicle applications. To overcome the out-of-distribution robustness issue, the models are required to be trained on a dataset that match the in-cabin sensor settings and environmental conditions (lighting, camera distance and image quality/resolution). In addition, the size and variability of this synthetic data may not be sufficient to learn descriptive features required in gaze estimation.

In an alternate evaluation, we assessed the effect of adding intermediate features, as explained in strategy 2 in previous section, to improve the performance of the gaze estimation models which are trained on synthetic data. The model proposed in strategy 2 (gaze estimation from landmarks) was trained on all images in the SynthesEyes dataset using the landmark annotations of 8 pupil counter points and the 3D gaze vector (Fig 4). As shown in Fig 8, the stacked Hourglass model has reasonable performance even under total occlusion in some eye images. The trained models are then tested using images of 15 subjects in the MPIIGaze dataset. The average mean angle error for all test images is reduced to 5.02 degrees which is comparable to ResNet18 trained on real data. This result shows the capability of training models synthetically in estimating the gaze angle which will reduce time and resources for real data collection and annotation process.

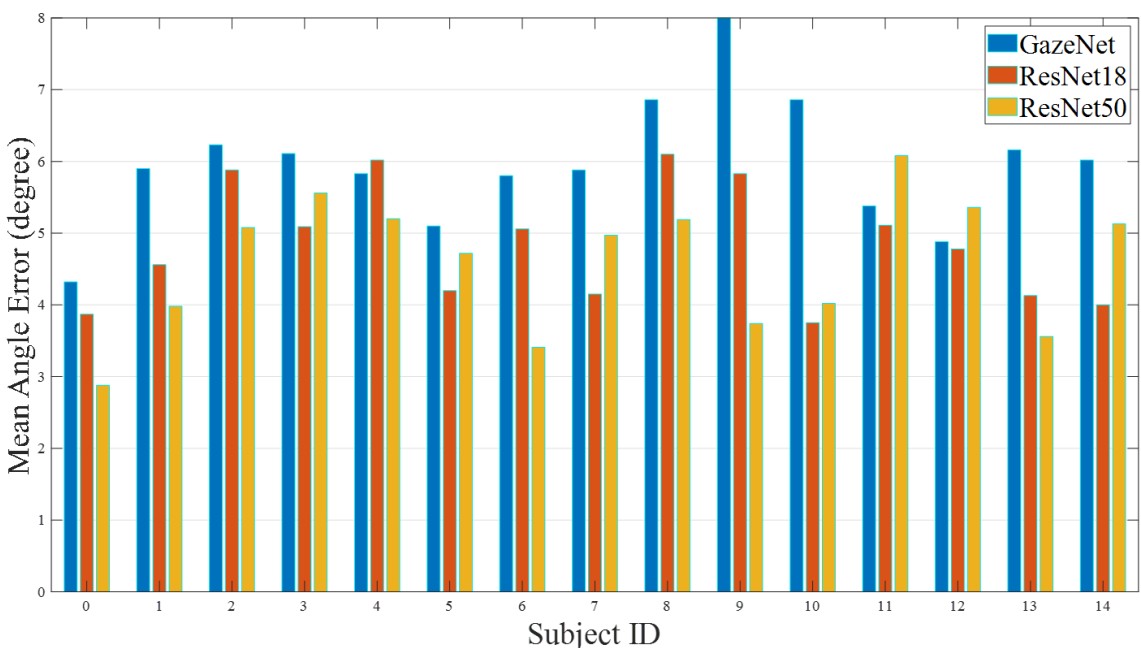

Figure 6: Angular errors of three models on the test subsets

Table 1: Average angular errors for testing three proposed models which are trained on real data and synthetic data

| Model | Test angle error[degree] (real training) | Test angle error[degree] (synthetic training) |
|---|---|---|
| GazeNet | 5.95 | 9.95 |
| ResNet18 | 4.83 | 7.90 |
| ResNet50 | 4.60 | 7.02 |

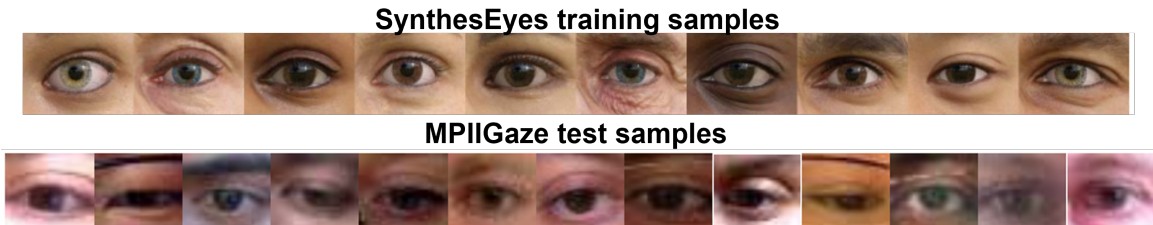

Figure 7: The appearance distribution among sample images in the real and synthetic datasets

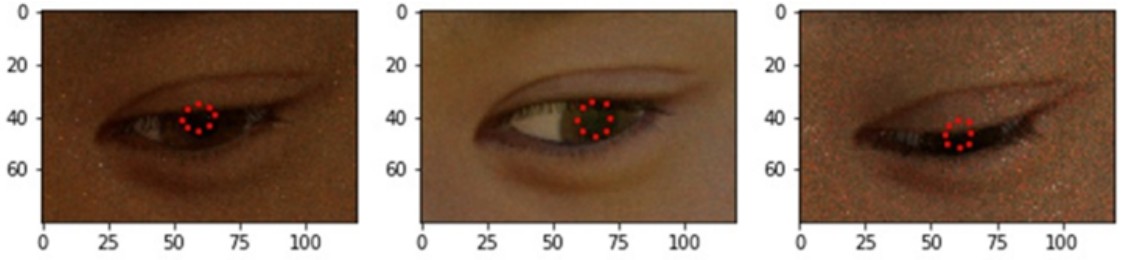

Figure 8: Examples of landmark detection results

## 6. The Influence of Evaluations Metrics

The state of the art methods, on gaze estimation, assess performance through common evaluation metrics, such as angular error, (in degrees), gaze recognition accuracy (in percentage) and location shifts between point of regards on a screen (in pixel/mm) Kar and Corcoran (2018). However, these metrics are very often defined based on special use cases such as relative to laptop screen or vehicle interior cockpit, and do not translate to contextually relevant interpretations such as in the case of a drivers gaze relative to attention on the road. Hence contextually relevant metrics are necessary to avoid ambiguities in performance assessment relative to specific requirements. For DSM features this is especially critical in reducing false positives/negatives related to drivers attention for optimal performance of $L_3$ TOR. If we define the TOR response time, as a function of predicted gaze angle, the scene in front of the ego vehicle and the vehicle driving parameters, thus, the gaze error will impact the final TOR response and therefore, the safety of $L_3$ functionalities. In this regard, the gaze angle error needs to be qualified in the context of $L_3$ driving. There is also a general lack of discussion on sensitivity analysis and clear interpretation of the effects of error ranges on display sizes and distances Kar and Corcoran (2018). For instance, in driver's attention monitoring and determining the zone of attention, assessment of the gaze angular (degrees) error, should be conducted relative to the on-the-road region boundaries with respect to the distance to the camera. In gaze estimation method, point of regard in 3D space is computed using the midpoint of the segment between estimated and true gaze vectors. Therefore, small angle errors cause large deviations in the depth of the point estimation. In Liu et al. (2020), the authors demonstrated that when the distance increases, gaze estimation achieves better performance, whereas in short distances the distance of the scene to camera becomes influential. This phenomena is shown in Fig 9.

## 7. Discussion and Conclusion

Gaze estimation is a powerful tool to measure human intent and visual attention in a wide range of applications. An intelligent interior vision system in automotive systems is essential to track the behavior of the driver in cabin, for driver safety and Level 3 hand-over functionalities. Driver's attention and point of regard can be measured by utilizing the estimated gaze angle from image data which is collected by interior camera. Appearance-based gaze regression approaches using CNN are presented in this paper, where human's gaze direction is estimated to measure where the human is looking. In this paper, the effectiveness

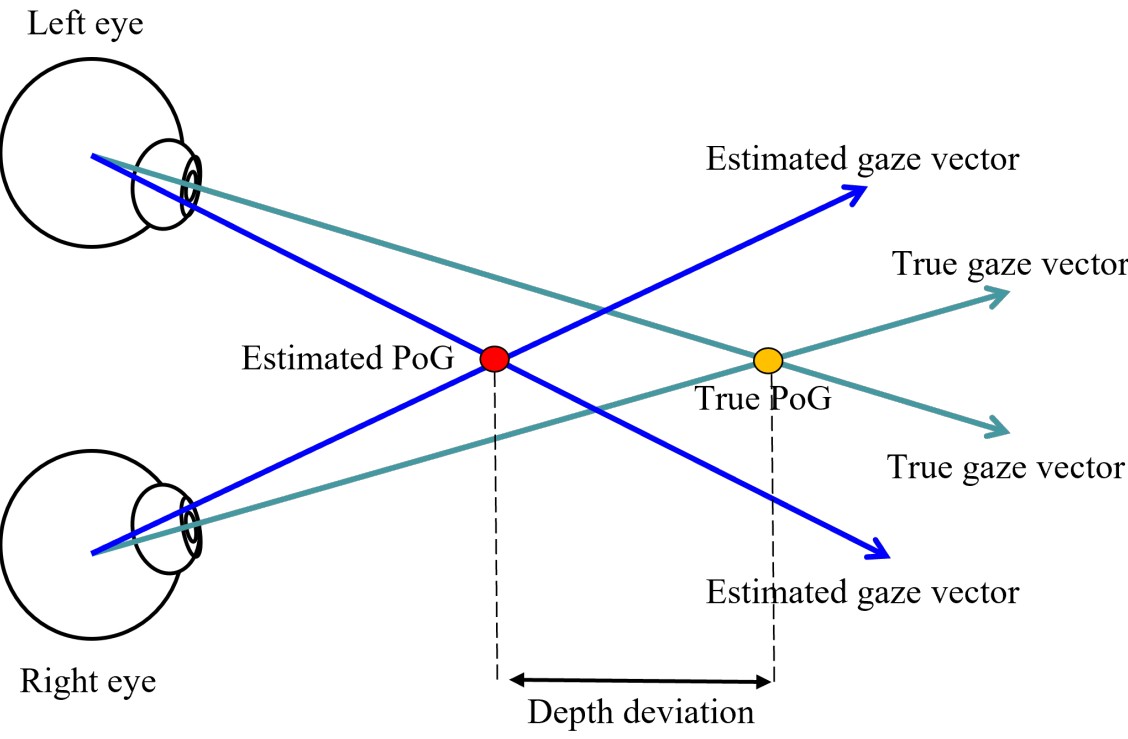

Figure 9: The effect of angular error on the distance estimation of point of regard Liu et al. (2020).

of various appearance-based gaze estimation approaches were studied. We evaluated the performance of DNN models by directly applying them to the eye image and regressing the gaze angle. A comparison was conducted between the models trained using the real data and synthetic data. Training models using synthetic data, not only takes advantage of accurate annotations which results in accurate representations, but also helps to deal with data need issue in deep learning. Due to the small size of the synthetic dataset and the lack of required appearance variations to generalize well to the real data distribution, synthetically learnt models' performance drastically dropped. We proposed landmark-based gaze estimation, to estimate gaze from eye features.That approach despite the learning synthetic representations, showed comparable results versus CNN models which were trained on real data.This model can be improved to be utilized for in-cabin gaze estimation in DSM by increasing the number of training samples and train on synthetic settings which match the in-cabin image settings and lighting conditions.We discussed that error metrics should be interpretable and be adapted to the applications more specifically, relative to the distance and scene settings, for an appropriate assessment.This phenomena will be investigated experimentally in the future using in-vehicle images.

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
