# OpenReview forum: "Appearance-Based Gaze Estimation for Driver Monitoring"
_NeurIPS.cc/2022/Workshop/GMML — Gaze Meets ML 2022 Poster_

### Official Review · Reviewer_Cuyn · 2022-10-08
**Appearance-Based Gaze Estimation for Driver Monitoring**

**Rating:** 4
**Confidence:** 4

**Review:**

The authors propose improving the training of the appearance-based gaze estimation for driver monitoring by (1) training on a synthetic data and (2) introducing an additional objective for predicting eye landmark coordinates.

Pros:
1. The approach is well motivated. The application of gaze estimation is important. The idea of training on a synthetic dataset and introducing a second objective to leverage annotations that are available in the synthetic data (eye landmark coordinates) and hard to obtain in real data is intuitive and plausible.

Cons:
1. The authors don't report results for strategy2. The results in table1 and figure6 are for strategy1. They only briefly mention one result for strategy2 in line 186.
2. Results for strategy1 seem to be pretty bad. The author didn't provide a good explanation. They initially argued that the synthetic dataset is realistic and more diverse than the real one. Also, they didn't show what happens if they finetune their model on the real data.
3. The authors didn't compare to all baselines in the field (cited in related work). They just compare to one baseline GazeNet and experiment with different architectures (Resnet18 and Resnet50)
4. typo: line7, beyond is repeated twice
5. The authors discuss the need for new metrics. They didn't provide examples from their experiments were current metrics drastically fail.

---

### Official Review · Reviewer_vd7F · 2022-10-17
**Interesting approach to train on synthetic data**

**Rating:** 7
**Confidence:** 4

**Review:**

Performance of different CNN architectures for regressing gaze angle from face images is compared. Further, the effect of training on real images and on synthetic images is compared. As expected, training on synthetic images leads to performance degradation when tested on real images, due to the distribution shift issue.

A method is proposed to reduce this performance drop, by introducing an intermediate task of detecting eye landmarks. The introduction of this intermediate task enables a model trained on synthetic images to have comparable performance to one of the models trained with real images.

Sec. 6 provides a discussion on which evaluation metrics are suitable for the gaze estimation task, keeping downstream applications in mind. I appreciate the initiation of this important discussion, but unfortunately, Sec. 6 provides only a vague discussion, without any concrete suggestions or experiments.

Overall strengths: methodological novelty (introduction on intermediate landmark detection task), initiation of discussion regarding application-specific evaluation metrics, good evaluation in a small dataset setting (n-fold cross validation, where n is the number of subjects).

I encourage authors to add standard deviation in table 1 and for the results of the model trained with the intermediate task.

---

### Official Review · Reviewer_Pa2F · 2022-10-19
**Good paper on use of synthetic data and landmarks, with unrelated driving context**

**Rating:** 7
**Confidence:** 3

**Review:**

Pros:
   The paper addresses an important general ML topic -- leveraging synthetic data -- and applies that topic to gaze estimation.
   The authors technique of augmenting the data based on the synthetic model is potentially very useful in wide variety of contexts.
   Paper is generally well written and easy to follow.
   Generally, this is a good paper in an important area; I criticize because I want it to be even better.

Things that would improve this paper:
(1) I'd like to see work on varying blends of real and synthetic data. How low of a ratio of real : synthetic still yields good results?
(2) A side by side illustration of the authors compared flows would help -- there are pieces such as landmarks and block diagram -- but no end to end flow for each method in similar presentation.
(3) I'd like to see a more detailed contrast with the the SynthesEyes dataset authors evaluation to establish originality.

Cons:
(1)  As far as I can tell, this is a paper on gaze angle estimation in a completely unrelated driving wrapper. Nothing I can find, other than intro and discussion, seems to be influenced by driving as context. It would be very interesting to see the study the authors suggest but do not do around making the evaluation metric driving specific. IMO, driving could be reduced to a paragraph on key application / future work and this would be a better paper.
(2) The paper does not discuss significance of errors. In particular, if the authors want to focus on a driving context (not recommended) -- does a few degrees matter at all? This could vary e.g. cursor movement vs. not looking at road?
(3) This seems like an application where even a few images over time would substantially improve results. Has the time dimensions been considered?

---

### Meta-Review · Area_Chair_hWns · 2022-10-20

**Recommendation:** Accept (Poster)
**Confidence:** 4

**Metareview:**

The paper describes a method for using synthetic data for gaze estimation. Authors augment with synthetic data and introduce an intermediate task to help improve the out-of-distribution performance to compensate for using the synthetic data for training. The eventual goal of this work is to use it for gaze monitoring in L3 self-driving cars.

The reviewers agree this is an important work combining synthetic data, especially in gaze estimation. Further, they also acknowledge the importance of application-specific metric discussion.

I recommend an acceptance and suggest the authors consider the reviewers' useful suggestions.

---

### Decision · Program_Chairs · 2022-10-20

Accept (Poster)